# Investigation of a Complex Reaction Pathway Network of Isobutane/2-Butene Alkylation by CGC–FID and CGC-MS-DS

**DOI:** 10.3390/molecules27206866

**Published:** 2022-10-13

**Authors:** Kaiwei Fu, Bei Liu, Xiaopeng Chen, Zhiyu Chen, Jiezhen Liang, Zhongyao Zhang, Linlin Wang

**Affiliations:** 1Guangxi Key Laboratory of Petrochemical Resources Processing and Process Intensification Technology, School of Chemistry and Chemical Engineering, Guangxi University, Nanning 530004, China; 2PetroChina Guangxi Tiandong Petrochemical Co., Ltd., Tiandong 531599, China

**Keywords:** capillary gas chromatography with flame ionization detection (CGC-FID), capillary gas chromatography-mass spectrometry-data system (CGC-MS-DS), alkylation, reaction network

## Abstract

The mechanism of reaction in isobutane/2-butene alkylation systems is extremely complicated, accompanied by numerous side reactions. Therefore, a comprehensive understanding of the reaction pathways in this system is essential for an in-depth discussion of the reaction mechanism and for improving the selectivity of the major products (clean fuel blend components). The alkylation of isobutane/2-butene was studied using a self-made intermittent reaction device with a metering, cooling, reaction, vacuum and analysis system. The alkylates were qualitatively and quantitatively analyzed using a capillary gas chromatography-mass spectrometry-data system (CGC-MS-DS) and capillary gas chromatography with flame ionization detection (CCGC-FID), respectively, and the precision and recovery of the quantitative analytical methods were verified. The results showed that the relative standard deviation (RSD) of the standard sample was below 0.78%, and the recoveries were from 98.53% to 102.85%. Under the specified reaction conditions, 79 volatile substances were identified from the alkylates, and the selectivity of C_8_ and trimethylpentanes (TMPs) reached 63.63% and 53.81%, respectively. The changes of the main chemical components in the alkylation reaction with time were tracked and analyzed, based on which reaction pathways were determined, and a complex reaction network containing the main products’ and the by-products’ generation pathway was constructed.

## 1. Introduction

With the increasing stricter emission standards, the upgrading of gasoline is developing in a higher quality and environmentally friendly direction. The alkylation of isobutane/butene in the C4-fraction is a vital process in the petroleum refining industry for the production of alkyl compounds, in which the alkylated oil produced has the outstanding advantages of a high researched octane, with good anti-explosive properties, no olefins or aromatics, a low vapor pressure, a complete combustion, cleanliness, etc. It is the unique blending component that can simultaneously satisfy the requirements of a high octane value and a clean combustion [1,2,3,4,5,6,7,8]. Since 1990, U.S. refiners have been obliged to change their gasoline composition strategies to meet the mandatory specifications of the Clean Air Act (CAA). Since 1995, the US has been using methyl tertiary butyl ether (MTBE) in large quantities for gasoline additives. However, due to the MTBE having a severe environmental contamination problem, policies banning MTBE as a gasoline additive have been introduced in many countries. Since then, the alkylates have become the optimum mixture component in the gasoline pool of a traditional refinery. With the rapidly increasing demand for alkylated oil, the world’s alkylated oil production exceeded 115.6 million tons/year [9,10,11,12].

Currently, liquid acids, ionic liquids and solid acids are used as catalysts for isobutanes and butenes alkylation reactions. However, alkylation technologies using ionic liquids and solid acids as catalysts are not yet mature. Moreover, they have high input costs, so these technologies have not yet been promoted in the industry. The commonly used catalysts for the industrial production of alkylated oils are still sulfuric acid and hydrofluoric acid, among which, the sulfuric acid method has the characteristics of being cheap and easy to obtain raw materials and mature technology; while hydrofluoric acid is a highly toxic, highly volatile and corrosive compound, which is extremely easy to volatilize and spread, causing ecological and environmental pollution, so sulfuric acid has been the mainstream alkylation catalyst in the industry. At present, there are about 700 refineries worldwide, and most of them are based on the sulfuric acid method [13,14,15,16,17].

The reaction pathway and the reaction kinetics are important to design and optimize the reaction process [18]. At the beginning of the 1940s, Schmerling [19,20] suggested a mechanism to describe the process of alkylation for olefins and isobutanes based on the ionic principle. Following that, Albright et al. [21,22] showed that the main source of dimethylhexanes (DMHs) carbonium ions was not generated by the isomerization of TMPs carbonium ions, as the amount of DMHs varied dramatically with the retention time and stirring speed. Sun et al. [23] developed a kinetic model for isobutane and butene alkylation reactions, where sulfuric acid was used as the catalyst, and the variation in the concentration of TMPs, DMHs and heavy ends (HEs) was predicted. Li et al. [14] used the microchemical system to determine the concentration variations in the sulfuric acid alkylation reaction of isobutane and 2-butene at different temperatures for the primary components of TMPs, DMHs, light ends (LEs) and HEs, and developed a simple kinetic model incorporating the main and side response reactions, which was well predicted and the results were further validated using COMSOL software simulations.

However, the C_4_ alkylation reaction is a complex reaction system, and its reaction process is accompanied by a variety of side reactions and generates a large number of by-products. To reduce the side reactions of the system, it is necessary to have a comprehensive understanding of the reaction pathways of this reaction system. Although some studies on alkylation reactions have been reported, few of the above-mentioned studies specifically considered each light and heavy component but rather assembled them into LEs and HEs, and little focus has been paid to how the reaction pathway network has been explored and constructed. The envisaged reaction pathway network diagrams were simpler and difficult to explain, in detail, the pathways through which side reactions occur, which requires more detailed reaction pathway network diagrams to describe alkylation reactions and refine the alkylation reaction mechanism, thus providing a basis for building a kinetic model, which in turn provides a basis for the design and optimization of reaction conditions and reactors. Meanwhile, to obtain a detailed alkylation reaction pathway network and investigate the reaction mechanism in-depth, the C_4_ alkylation reaction system with sulfuric acid as the catalyst must be traced and analyzed in detail to comprehensively identify and quantitatively analyze the reaction species at different reaction times.

CGC-FID is a sensitive, accurate, reproducible, quantitative and versatile analytical tool that is well suited for analyzing complex mixtures. The CGC–MS-DS is one of the most attractive and effective means among the commonly used qualitative analysis methods due to its good sensitivity, high selectivity and versatility, as well as a large number of well-established library databases available [24,25,26,27,28,29]. In this study, experiments on the alkylation of C4-fractions with sulfuric acid as the catalyst were carried out by a self-designed apparatus with a metering system, a cooling system, a reaction system and an analysis system. The CGC-MS-DS was used to identify a complex mixture of alkylation products. The quantitative analysis was performed using CGC-FID, and this quantitative method’s calibration factors, precision and spiked recoveries were investigated. Meanwhile, the intermediate species and products distribution were investigated by the CGC–MS-DS and CGC-FID. According to the classical carbonium ion mechanism, as well as the results of the CGC-MS-DS identification of the alkylation products of isobutane and 2-butene and the alkylation product distribution with time, a detailed reaction pathway network was developed, which includes the reaction pathway of each by-product.

## 2. Results and Discussion

### 2.1. Optimization of the CGC-FID Analysis Conditions

Since almost all alkyl compounds are hydrocarbons, the inlet temperature was set up at 250 °C to rapidly vaporize the various components of the sample. The temperature of the detector was set up at 250 °C to prevent the generation of condensate in the detector, due to the temperature difference. The alkylates are a mixture of hydrocarbons with different carbon numbers, in which there are a large number of isomers with similar boiling points and molecular weights and wide boiling ranges, so the CGC separation was carried out by a programmed temperature rise. Following several experimental tests on the injection volume, the programmed ramp-up rate and the carrier gas flow rate, the CGC-FID analysis conditions were identified below: both the injector and flame ionization detector were set to 250 °C, and the injection was in separation mode (1:80) with an injection volume of 1 μL. The pressures of the carrier gas, air, and hydrogen were set to 0.12 MPa, 0.1 MPa, and 0.1 MPa, respectively. This analytical condition provided a good separation of the alkylate fractions, and the chromatogram of the alkylates is shown in Figure 1.

### 2.2. Quantitative Analysis Results of the Peak Area Normalization Method

#### 2.2.1. Relative Correction Factors

Three standard solutions of different concentrations were prepared separately using electronic-analytical balances with a precision as low as 0.0001 g. The standard solutions were measured five times in parallel under the specified chromatographic analysis conditions, and 2,2,4-trimethylpentane was selected as the reference material (S). The relative correction factors of each component were calculated by Equation (4), as shown in Table 1.

As seen in Table 1, the relative correction factors of the benchmark 2,2,4-trimethylpentane and the components were in the range of 0.99 to 1.07, and all of the components had relative correction factors close to 1. For the alkylates, most of the components are tautomers or homologs, and their structures are similar, so their relative correction factors are also close to each other.

#### 2.2.2. Comparison of the Peak Area Normalization Method and the Corrected Peak Area Normalization Method

Firstly, three standard samples with different concentrations were prepared and quantified by the peak area normalization method and the corrected peak area normalization method. Next, the average value was taken for five parallel measurements, and then the relative deviations (RDs) were calculated according to Equation (1), and the results are shown in Table 2.
(1)$$RD=\frac{Z_{i}-v_{i}}{v_{i}}\times 100\%$$
where *Z_i_* is the concentration of component *i* measured by different analytical methods, *v_i_* is the concentration of the initial standard samples of component *i*.

As seen in Table 2, the maximum RD between the quantitative results of the peak area normalization method and the actual content is 3.11%, and the maximum RD between the quantitative results of the corrected peak area normalization method and the actual content of the sample is 2.52%. The difference between the quantitative results of the two ways was 0.59%. The deviation of the data measured by the two methods is small, so it is more convenient to choose the peak area normalization method for the quantification.

#### 2.2.3. Precision

A standard sample was prepared, measured five times in parallel and the average value was calculated. Then, the measurement results’ relative standard deviation (RSD) was calculated by Equation (2), and the results are shown in Table 3.
(2)$$RSD=\frac{S_{i}}{{\bar{x}}_{i}}=\frac{\sqrt{\sum_{j=1}^{n}{\left(x_{ij}-{\bar{x}}_{i}\right)}^{2}}}{{\bar{x}}_{i}}\times 100\%$$
where *S*_i_ is the standard deviation of component *i*, *x*_ij_ is the mass fraction of component i, *x*_i_ is the mean of the mass fraction of component *i*.

As shown in Table 3, the RSD of the samples was less than 0.78%, indicating that the selected chromatographic conditions were reasonable and the excellent precision of the area normalization method’s quantitative results met the determination requirements.

#### 2.2.4. Recovery

A standard sample was prepared and measured five times in parallel to take the average value, and then a specific content of the measured substance was added and measured five times in parallel by the same method. Finally, the recoveries of the components in the samples were calculated according to Equation (3), and the results are shown in Table 4.

As shown in Table 4, the recoveries of the standard samples were 98.53% to 102.85%, and the accuracy of the selected analytical and quantitative methods was high.
(3)$$Recovery=\frac{Y_{i}-y_{i}}{a_{i}}\times 100\%$$
where *Y_i_* is the measured value after the spiking of component *i*, *y_i_* is the measured value of the initial standard sample of component *i*, *a_i_* is the spiked value of component *i*.

### 2.3. Chemical Composition of the C_4_ Alkylation Products

The alkylates were characterized using the CGC-MS-DS. The standard samples (listed in the materials) were first analyzed to determine their spectrum and relative retention time, and then the alkylate was divided into two equal parts. One was added to the standard samples, and the other was not added with any substance. Both samples were analyzed using identical instrument parameters. Their spectra were both searched using the NIST14, NIST14s, NIST20-1, NIST20-2 and NIST20s databases in the CGC-MS-DS program. Then, the substances without standard samples were directly searched and characterized by the databases, and other products in the alkylate were characterized by the databases, further confirmed by adding standards. A total of 86 substances were confirmed to be isolated from the isobutane/2-butene alkylation reaction products, and the identification of 79 compounds was identified. The results revealed that the retention times of the most important products of the isobutane/2-butene alkylation reaction, TMPs, ranged from 16.734 min to 22.407 min. The results of the isobutane/2-butene alkylation product components are shown in Table 5.

### 2.4. Alkylation Reaction Pathway Network

#### 2.4.1. Changes in the Composition of the Reaction Process

The isobutane/2-butene alkylation reaction was followed by the CGC–MS-DS and CGC-FID under the conditions of the molar ratio of isobutane to 2-butene of 10:1 (I/O = 10:1), the sulfuric acid/hydrocarbon volume ratio of 1:1 (A/H = 1:1), the reaction temperature of 7 °C, the reaction pressure of 0.5 MPa, the stirring speed of 1300 rpm and the changes of the reaction conversion and selectivity of each component with time were investigated. The results are shown in Figure 2.

As noted in Figure 2d, the conversion of 2-butene reached 97.12% at 2 min, which indicates that the alkylation of C_4_ was a fast reaction, and most of 2-butene was already consumed rapidly at 2 min, and the conversion of 2-butene increased slightly after 2 min; it reached 98.08% at 5 min and finally stabilized, but the conversion did not reach 100%, perhaps because 2-butene had a certain saturation vapor pressure at the reaction temperature and failed to participate in the reaction completely. From Figure 2a–c, it could be seen that the selectivity of the C_8_ component increased sharply from 0.5 to 5 min, the selectivity of the C_9_^+^ components decreased significantly, and the selectivity of the C_5_-C_7_ components increased slowly; after 5 min, the components stabilized. The alkylation reaction was completed after 2 min, but between 2 and 5 min, the selectivity of the TMPs continued to increase, the selectivity of the DMHs increased slightly, and the selectivity of the C_9_^+^ high carbon fraction continued to decrease.

#### 2.4.2. Reaction Pathway Network Construction

According to the CGC–MS-DS and CGC-FID tracing analysis results of the alkylation reaction products, it was known that the reaction generated C_8_ (TMPs) as the main product, while a large number of low carbon molecules, as well as high carbon molecule by-products, were also generated, in which the high carbon molecules were generated because of the polymerization of the low carbon molecules. In the qualitative analysis of the CGC–MS-DS in Table 5, it was found that there were many isomers in the same carbon number molecule, indicating that there was also an isomerization reaction in the alkylation reaction. It could be seen from Figure 2 that the alkylation reaction essentially ended at 2 min, and within 5 min, the selectivity of C_8_ increased sharply, the selectivity of C_9_^+^ decreased sharply, and the selectivity of C_5_–C_7_ increased slightly. Indicating that at this stage, high carbon molecules underwent a scission reaction to form C_8_ and the low carbon molecules C_5_–C_7_, among which TMPs are the main cleavage products. The selectivity of C_9_ decreased most rapidly, indicating that it was one of the significant reactants in the fragmentation reaction. Thus, the alkylation of isobutene/2-butene was a complex reaction in which the primary reaction was an addition reaction to produce C_8_, accompanied by polymerization, fragmentation, isomerization and other side reactions. The reactions at each node in the alkylation reaction pathway network were as follows:

(1) Isomerization reaction: Under an acidic environment, the reaction material 2-butene (2-C_4_^=^) undergoes isomerization through a hydrogen transfer or methyl transfer to form 1-butene (1-C_4_^=^) and isobutene (i-C_4_^=^) and reaches equilibrium, and the thermodynamic equilibrium between butenes favors isobutene at the reaction temperature, so the selectivity of isobutene is the highest [30,31]. Similarly, other high-carbon carbocations undergo isomerization reactions through a similar hydrogen transfer or methyl transfer, which is why the alkylation products contain multiple isomers in the same carbon number molecule.

(2) Main reaction: It is well known that the isobutane alkylation reaction follows the classic carbonium ion mechanism by Schmerling et al. [18,32]. According to the classic carbonium ion mechanism, the unsaturated double bond in butene seizes H^+^ in the acid catalyst to form C_4_^+^, and C_4_^+^ further undergoes isomerization to generate the more stable tert-butyl carbocation (i-C_4_^+^). i-C_4_^+^ underwent addition reactions with 2-C_4_^=^ (or i-C_4_^=^) and 1-butene to generate TMPs^+^ and DMHs^+^, respectively. Finally, the TMPs^+^ seizure the H- of i-butane (i-C_4_) to generate TMPs^+^ and the DMHs^+^ seizure the H^-^ of reactants i-C_4_ to generate DMHs [14,33].

(3) Polymerization reaction: Olefins underwent dimerization or multimerization reactions at high temperatures and under acidic conditions. The strong exotherm of the alkylation reaction led to high local temperatures and initiated the polymerization of olefins. The dimerization reaction between olefins produced C_8_^+^, and then C_8_^+^ and short chain carbonium ions will continue to polymerize with C_4_-fractions to produce high carbon number molecules [34,35].

(4) Fragmentation reaction: The multimerization reaction between olefins generated high-carbon molecules, which obtained protons to form carbonium ions. Long-chain carbonium ions were unstable in an acidic environment and were easily broken into short-chain hydrocarbon molecules at the β position of the charged carbon atoms. The resulting short-chain hydrocarbons underwent further reactions under alkylation conditions [36]. Albright [37] believed that C_12_^+^ and C_16_^+^ were the most important intermediates, C_12_^+^ and C_16_^+^ underwent a cleavage reaction to generate short-chain carbanions and short-chain alkenes, and the short-chain carbanions further abstracted H^-^ to generate short-chain alkanes, resulting in the generation of C_5_, C_6_, C_7_ and other alkanes.

According to the classic carbonium ion mechanism, the results of the tracing analysis of the isobutane/2-butene alkylation products, and the discussion of the above reaction pathway network nodes, multiple reactions occur simultaneously in the alkylation system of isobutane and 2-butene with a large number of isoparaffins and corresponding carbocations. During the experiment, it was found that the change in the selectivity of the C_8_ component was negatively correlated with the selectivity of C_9_^+^ components before 5 min, with the largest change in the selectivity of the C_9_ components. However, as shown by the conversion of 2-butene, the alkylation reaction was finished at 2 min. Therefore, the increase in the selectivity of the C_8_ components between 2 to 5 min may be related to the fracture reaction of the C_9_ components. In this work, an isobutane/2-butene alkylation reaction pathway network with main and side reactions was constructed on the basis of the results obtained from the tracing analysis, as shown in Figure 3.

## 3. Experimental Section

### 3.1. Materials

2,3-dimethylbutane, 2-methylpentane, 3-methylpentane, 2,4-dimethylpentane, 2,2,3-trimethylbutane, 2-methyl Hexane, 2,3-dimethylpentane, 3-methylhexane, 2,2,4-trimethylpentane, 2,4-dimethylhexane, 2,2,3-trimethyl pentane, 2,3,4-trimethylpentane, 2,3-dimethylhexane, 3,4-dimethylhexane and 2,2,5-trimethylhexane were all certified standard samples (chromatographic grade), purchased from TCI (Shanghai) Huacheng Industrial Development Co., Ltd. (Shanghai, China) without further treatment. H_2_SO_4_ (AR 96–98%) was purchased from Chengdu Kelon Chemical Co., Ltd. The high-purity gases such as hydrogen (H_2_), nitrogen (N_2_), helium (He), dry air, isobutane and 2-butene were purchased from Guangdong Huate Gas Co., Ltd. (Nanning, China). The raw material of isobutane and 2-butene is a mixture, and the ratio of isobutane to 2-butene is 10:1 (I/O = 10:1, mol:mol).

### 3.2. Experimental Equipment

An independently designed set of C_4_ alkylation batch reaction devices, which consists of a metering system, a cooling system, a reaction system, a vacuum system and an analysis system. The feeding and metering of the raw materials are controlled by the plunger-type metering pump into the pressurized reaction kettle, but the control of the feed by the plunger-type metering pump may cause an inaccurate metering [38]. To solve the problem of inaccurate measurement, a mass flow meter (D07-7B, Seven Star, Beijing, China) was used to accurately measure the raw material gas. Then the feed gas passed through a constant temperature circulating water tank into a cooling coil, where it condensed and finally passed into a stainless steel PTFE-lined autoclave (Dalian Jingyi Autoclave Co., Ltd., Dalian, China) to perform the reaction. The unit is also equipped with a vacuum system to remove the air in the system so that the feed gas can be better condensed in the cooling coil and provide a stable environment for the alkylation. To investigate the variation of the reaction product concentration distribution with time, sampling is required to follow the reaction process. Therefore, the analytical system in this paper uses the CGC–MS-DS (TQ8040, Shimadzu, Kyoto, Japan) and CGC-FID (GC9790, FULI, Hangzhou, China) to perform the qualitative and quantitative analyses for the alkylation products, respectively. The self-made reaction device is shown in Figure 4.

### 3.3. Quantitative Analysis

#### 3.3.1. CGC-FID Analysis Conditions

Quantitative analysis was detected by FULI GC-9790 (Zhejiang Fuli Analytical Instruments Co., Hangzhou, China)with a FID detector and a HP-PONA capillary column (50 m × 0.2 mm × 0.5 μm). The temperature of both the injector and the flame ionization detector was set to 250 °C. The injections were made in a split mode (1:80) with an injection volume of 1 μL. The pressures of the carrier gas, air and hydrogen were set to 0.12 MPa, 0.1 MPa and 0.1 MPa, respectively. The temperature program was as follows: The column temperature was stabilized at an initial value of 60 °C (held for 1 min), ramped up to 80 °C (held for 2 min at a rate of 5 °C/min) and finally ramped up to 200 °C (held for 10 min).

#### 3.3.2. Determination of the Relative Correction Factors

According to the benchmark of a particular component, it was necessary to configure the standard solutions of the different concentrations. Next, we performed the parallel determinations under the specified chromatographic conditions. Then, based on the obtained data, we calculated the relative correction factors of other components in the solution relative to the reference substance.

The calculation formula is shown as follows:(4)$$f_{is}=\frac{f_{i}}{f_{s}}=\frac{m_{i}/A_{i}}{m_{i}/A_{i}}$$
where *f_is_* is the relative correction factor of component *i*, *f_i_* is the correction factor of component *i*, *f_s_* is the correction factor of a reference substance, *m_i_* is the mass of component *i*, *m_s_* is the mass of the reference substance, *A_i_* is the peak area of component *i*, *A_s_* is the peak area of the reference substance.

#### 3.3.3. Determining the Mass% of the SAMPLE Components

Based on the peak area Ai and the relative correction factor *f_is_* of component *i* concerning the reference material calculated in Equation (4), the mass fraction of each component *i* can be calculated from Equation (5).

The calculation formula is shown as follows:(5)$$\omega_{i}=\frac{A_{i}f_{is}}{\sum A_{i}f_{is}}\times 100\%$$
where *A_i_f_is_* is the corrected peak area of component *i*, ∑*A_i_f_is_* is the sum of the corrected peak areas, and *ω_i_* is the mass% of component *i*.

### 3.4. Qualitative Analysis

#### CGC-MS-DS Analysis Conditions

The instrument used for the qualitative analysis was a Shimadzu GC-MS-TQ8040, and the column installed was a HP-PONA (50 m × 0.2 mm × 0.5 μm) from Agilent, USA. The sample was injected in the split-flow model (1:100) with an injection volume of 0.2 μL. The column temperature was stabilized at an initial value of 60 °C (held for 2 min), ramped up to 80 °C (held for 2 min at a rate of 5 °C/min) and finally ramped up to 200 °C (held for 10 min). A high purity helium was used as the carrier gas with a pressure of 0.12 MPa, a total flow rate of 50 mL/min, and a purge flow rate of 3.0 mL/min. The total run time was 58 min. The inlet temperature was 250 °C. The mass spectra were obtained in a EI (electron ionization) mode at 70 eV. The ion source temperature was 250 °C, and the interface temperature was 280 °C. Full scan chromatograms with selected mass-to-charge ratios in the range of 20–300 *m*/*z* were used for the acquisition.

### 3.5. C_4_ Alkylation Reaction

The isobutane/2-butene alkylation was carried out in a 0.1 L stainless steel PTFE-lined autoclave. An internal water-cooled coil is used to control the reaction temperature. A schematic of the experimental set-up is depicted in Figure 4. A certain amount of concentrated sulfuric acid catalyst was poured into the autoclave, and then closed and sealed the autoclave. The air was extracted out of the autoclave with a vacuum pump to an absolute pressure of approximately 0.005 MPa. Then, N_2_ was introduced into the autoclave to bring the pressure to 0.5 MPa and held for 10 min to ensure that the autoclave did not leak. Next, the autoclave was purged three times by N_2_ at 0.5 MPa to eliminate any remnant air. The cryostat was adjusted to keep the cooling coil at a low temperature, and the material gas entered the cooling coil to condense. When the autoclave contents were cooled to the desired temperature, N_2_ was started to be charged to press the condensed material in the cooling coil into the autoclave while the alkylation reaction was carried out at a preset stirring rate. N_2_ was charged through a manually controlled valve to ensure a constant pressure in the autoclave throughout the reaction. Online sampling was performed at the desired time points, and the samples were processed before the gas chromatography analysis. Once the reaction was finished, we removed the product after collecting the gas from the autoclave.

## 4. Conclusions

The method for the CGC-FID analysis of alkylates was established, and the relative correction factors of several vital components in the alkylated gasoline were in the range of 0.9907 to 1.0692, and the maximum error of the quantitative results was obtained by the peak area normalization method and the corrected peak area normalization method was 0.59%. In addition, the precision and recovery of the area normalization method were examined. The results showed that the relative standard deviations of the precision were less than 0.78%, and the recoveries ranged from 98.53% to 102.85%, which indicated that the selected gas chromatographic conditions were reasonable and the quantitative analysis results by the area normalization method and the precision met the requirements of the assay.

The products of the alkylation reaction of isobutane/2-butene were characterized by the CGC–MS-DS coupling technique and a total of 79 compounds were identified and followed up. In the early stage of the reaction, 2-butene was isomerized in an acidic environment to form isobutene and 1-butene. In the first 2 min, isobutane was alkylated with the isomerized butene to form the C_8_ component (TMPs) and other by-products (C_5_–C_7_, C_9_^+^); after 2 min, the alkylation reaction was completed, and the rearrangement between the components of the reaction products was carried out between 2 and 5 min, with the long chain The long-chain alkane component undergoes a breakage reaction and the short-chain alkane undergoes an isomerization reaction; after 5 min, the rearrangement reaction process is completed. A detailed network of the alkylation of isobutane/2-butene containing each of the by-products was established, and the primary reaction is the alkylation of isobutene and 2-butene, which is a fast reaction accompanied by side reactions such as isomerization, polymerization, fragmentation, etc.

This study provides a precise and sensitive method of qualitative identification and quantitative analysis for the C_4_ alkylation, demonstrates a plausible complex reaction pathway network of alkylation to reveal the alkylation reaction mechanism, and provides a basis for the establishment of the detailed kinetic models, which in turn leads to the optimization reaction conditions and the design of the C_4_ alkylation reactors.

## Figures and Tables

**Figure 1 molecules-27-06866-f001:**
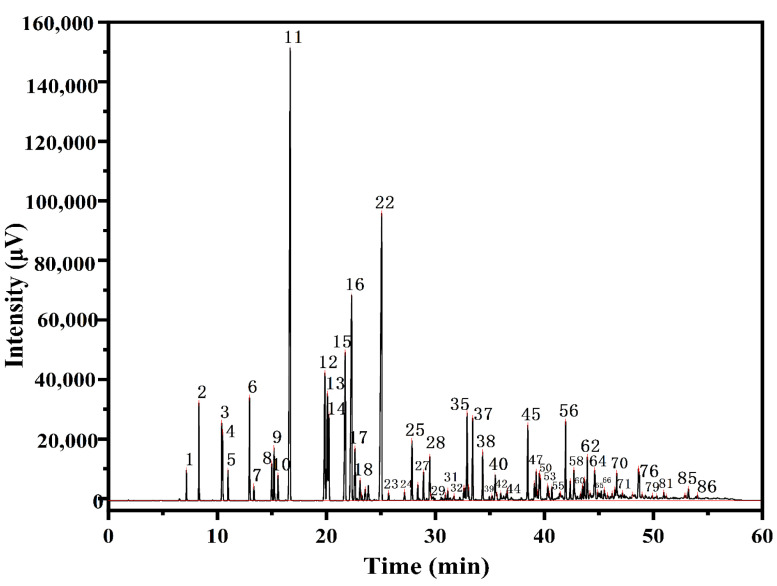
Gas chromatogram of the C_4_ alkylation reaction products.

**Figure 2 molecules-27-06866-f002:**
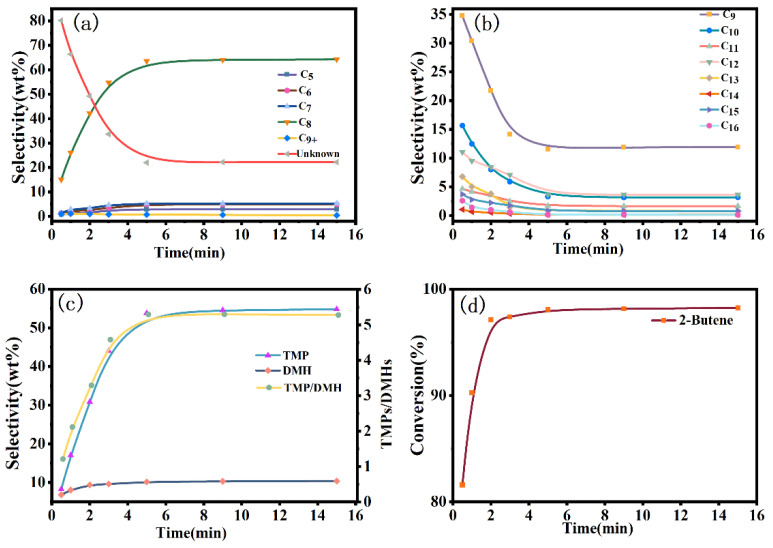
(**a**) Variation of the reaction product selectivity with time. (**b**) Variation of the C_9_^+^ selectivity with time. (**c**) Variation of the C_8_ and TMPs/DMHs values with time. (**d**) Variation of the 2-butene conversion with time. (H_2_SO_4_ volume = 30 mL, I/O = 10:1, A/H = 1.0, T = 7 °C, stirring speed = 1300 rpm).

**Figure 3 molecules-27-06866-f003:**
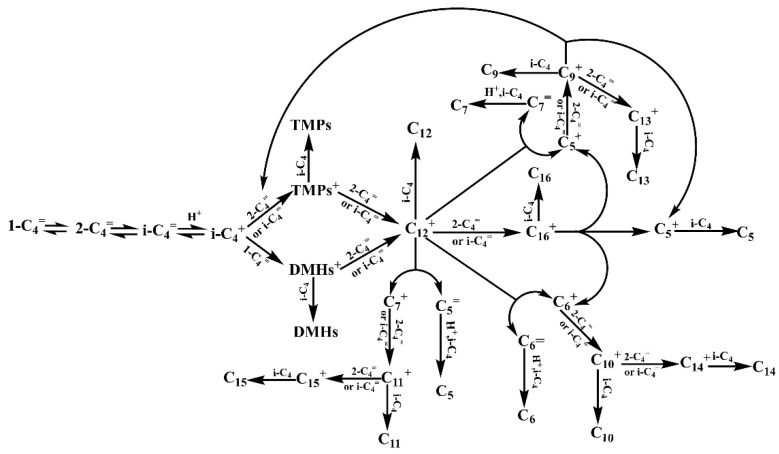
Diagram of the isobutane/2-butene alkylation reaction pathway network.

**Figure 4 molecules-27-06866-f004:**
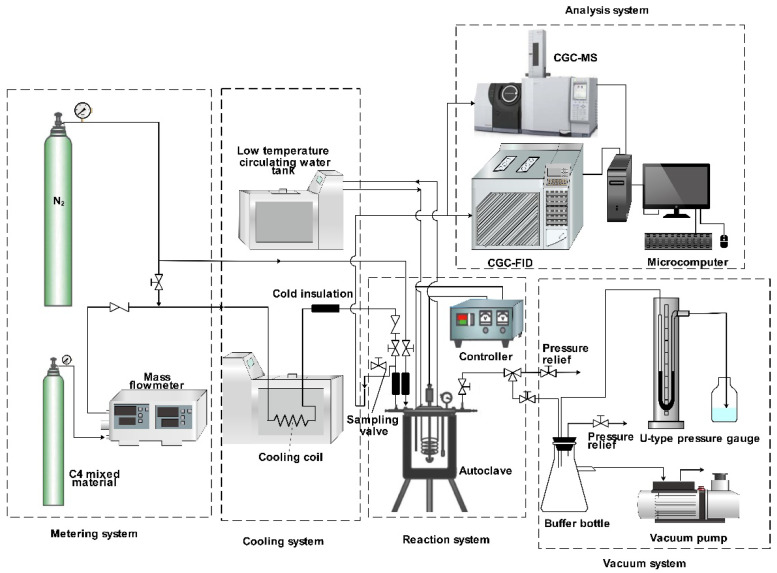
Diagram of an experimental device for the C_4_ alkylation reaction.

**Table 1 molecules-27-06866-t001:** Relative correction factors for the components.

Component	Relative Correction Factors (f_is_)			
	1	2	3	Mean Value
2,3-Dimethylbutane	1.0381	1.0243	1.0040	1.02
2-Methylpentane	1.0193	1.0236	1.0165	1.02
3-Methylpentane	1.0121	0.9946	1.0057	1.00
2,4-Dimethylpentane	1.0048	0.9717	1.0044	0.99
2,2,3-Trimethylbutane	1.0402	1.0541	1.0479	1.05
2-Methylhexane	0.9846	0.9856	1.0098	0.99
2,3-Dimethylpentane	0.9939	0.9943	0.9846	0.99
3-Methylhexane	1.0581	1.0351	1.0566	1.05
2,2,4-Trimethylpentane	1.0000	1.0000	1.0000	1.00
2,4-Dimethylhexane	1.0022	1.0076	0.9938	1.00
2,2,3-Trimethylpentane	1.0070	1.0465	1.0610	1.04
2,3,4-Trimethylpentane	1.0074	1.0017	1.0399	1.02
2,3-Dimethylhexane	1.0389	1.0839	1.0848	1.07
3,4-Dimethylhexane	1.0013	1.0356	1.0518	1.03
2,2,5-Trimethylhexane	0.9701	0.9923	1.0096	0.99

**Table 2 molecules-27-06866-t002:** Comparison of the quantitative analysis results of the peak area normalization method and the corrected peak area normalization method.

Component	Serial Number	Initial Concentration (wt%)	Concentration Measurement for the Quantitative Analytical Method			
			Peak Area Normalization (wt%)	RD (%)	Corrected Peak Area Normalization (wt%)	RD (%)
2,3-Dimethylbutane	1	3.5872	3.5260	1.71	3.5563	1.02
	2	8.0537	7.9519	1.26	8.0048	0.61
	3	4.4130	4.3533	1.35	4.3821	0.70
2-Methylpentane	1	5.6946	5.6813	0.23	5.7167	0.15
	2	6.6314	6.6641	0.49	6.6959	0.93
	3	5.2425	5.1777	1.24	5.1999	0.81
3-Methylpentane	1	7.1512	7.0880	0.88	7.0226	1.99
	2	5.0339	5.0358	0.04	5.0005	1.02
	3	6.7699	6.7055	0.95	6.6308	2.06
2,4-Dimethylpentane	1	4.9740	5.0375	1.28	4.9389	0.85
	2	5.9555	5.9559	0.01	5.8615	2.14
	3	8.5898	8.6865	1.13	8.4999	1.05
2,2,3-Trimethylbutane	1	4.1528	4.1006	1.26	4.2379	1.94
	2	5.1249	5.0591	1.28	5.1933	1.83
	3	8.1348	8.0010	1.64	8.2529	1.45
2-Methylhexane	1	7.1202	7.1621	0.59	7.0197	1.49
	2	4.3302	4.4232	2.15	4.3520	0.08
	3	6.4961	6.6022	1.63	6.4584	0.58
2,3-Dimethylpentane	1	7.0737	7.1637	1.27	7.0045	1.04
	2	8.3446	8.4279	1.00	8.2767	1.44
	3	7.5703	7.7102	1.85	7.5242	0.61
3-Methylhexane	1	5.3537	5.2108	2.67	5.3983	0.78
	2	9.7008	9.5368	1.69	9.8088	1.65
	3	8.3076	8.1247	2.20	8.4007	1.12
2,2,4-Trimethylpentane	1	13.6205	13.8235	1.49	13.6397	0.17
	2	9.7481	9.8102	0.64	9.7042	0.89
	3	11.6904	11.8166	1.08	11.6369	0.46
2,4-Dimethylhexane	1	7.5308	7.7190	2.50	7.6254	0.95
	2	12.0867	12.1274	0.34	12.0074	1.07
	3	5.2154	5.3102	1.82	5.2357	0.39
2,2,3-Trimethylpentane	1	7.0814	6.9541	1.80	7.1235	1.17
	2	3.9665	3.9703	0.10	4.0466	2.34
	3	3.6356	3.5926	1.18	3.6729	1.03
2,3,4-Trimethylpentane	1	7.9879	7.9515	0.14	7.9739	0.06
	2	5.3991	5.4484	0.91	5.4595	1.00
	3	5.9442	5.9357	0.46	5.9409	0.04
2,3-Dimethylhexane	1	4.3620	4.2262	3.11	4.4585	2.30
	2	4.3818	4.2664	2.63	4.4528	2.52
	3	5.9779	5.8033	2.92	6.1105	2.22
3,4-Dimethylhexane	1	6.5313	6.5465	0.28	6.6504	1.10
	2	5.0972	5.0845	0.25	5.1479	1.14
	3	4.4632	4.4506	0.23	4.5125	1.95
2,2,5-Trimethylhexane	1	7.7787	7.8093	0.39	7.6336	1.69
	2	6.1456	6.2382	1.51	6.1248	0.97
	3	7.5493	7.7298	2.39	7.5412	0.11

**Table 3 molecules-27-06866-t003:** Precision measurement results.

Component	Measured Value (wt%)					Mean Value (wt%)	RSD (%)
	1	2	3	4	5		
2,3-Dimethylbutane	4.7289	4.6799	4.6630	4.6613	4.6298	4.6726	0.78
2-Methylpentane	7.7475	7.6278	7.6606	7.6680	7.6258	7.6659	0.64
3-Methylpentane	3.5913	3.5486	3.5586	3.5540	3.5390	3.5583	0.56
2,4-Dimethylpentane	6.2783	6.2489	6.2517	6.2546	6.2558	6.2578	0.19
2,2,3-Trimethylbutane	3.9137	3.8916	3.9033	3.9071	3.9019	3.9035	0.21
2-Methylhexane	6.8789	6.8749	6.8655	6.8750	6.8775	6.8743	0.08
2,3-Dimethylpentane	6.4327	6.4289	6.4326	6.4426	6.4411	6.4356	0.09
3-Methylhexane	10.9570	10.9688	10.9536	10.9686	10.9730	10.9642	0.08
2,2,4-Trimethylpentane	23.2735	23.3669	23.3195	23.3433	23.3566	23.3320	0.16
2,4-Dimethylhexane	4.3548	4.3370	4.3977	4.3747	4.3748	4.3678	0.53
2,2,3-Trimethylpentane	2.9461	2.9966	2.9515	2.9656	2.9797	2.9679	0.70
2,3,4-Trimethylpentane	5.8892	5.9298	5.9284	5.9132	5.9313	5.9184	0.30
2,3-Dimethylhexane	6.1666	6.2031	6.2007	6.1866	6.2028	6.1919	0.25
3,4-Dimethylhexane	2.6111	2.6291	2.6386	2.6291	2.6371	2.6290	0.42
2,2,5-Trimethylhexane	4.2305	4.2682	4.2747	4.2564	4.2741	4.2608	0.43

**Table 4 molecules-27-06866-t004:** Determination results of the recovery.

Component	Initial Value (g)	Spiked Value (g)	Measured Value (g)	Recovery (%)
2,3-Dimethylbutane	0.1488	0.1091	0.2563	98.53
2-Methylpentane	0.0926	0.0637	0.1555	98.74
3-Methylpentane	0.0933	0.0946	0.1868	98.76
2,4-Dimethylpentane	0.1085	0.1571	0.2688	102.05
2,2,3-Trimethylbutane	0.1404	0.0613	0.2012	99.22
2-Methylhexane	0.1179	0.1320	0.2498	99.86
2,3-Dimethylpentane	0.1277	0.1836	0.3129	100.85
3-Methylhexane	0.1163	0.1526	0.2670	98.76
2,2,4-Trimethylpentane	0.1916	0.1208	0.3136	100.99
2,4-Dimethylhexane	0.0746	0.0814	0.1584	102.85
2,2,3-Trimethylpentane	0.0553	0.0277	0.0836	101.97
2,3,4-Trimethylpentane	0.0883	0.0773	0.1647	98.86
2,3-Dimethylhexane	0.0759	0.0755	0.1505	98.77
3,4-Dimethylhexane	0.1003	0.0739	0.1731	98.53
2,2,5-Trimethylhexane	0.0895	0.0928	0.1825	100.20

**Table 5 molecules-27-06866-t005:** CGC-MS-DS analysis results of the alkylation products.

Serial Number	Retention Time (min)	Peak Area (%)	Similarity (%)	Alkylate Components	Molecular Weight	Molecular Formula
1	7.150	0.4114	95	Isobutane	58	C_4_H_10_
2	8.292	1.4791	88	2-Methylbutane	72	C_5_H_12_
3	10.392	1.3210	90	2,3-Dimethylbutane	86	C_6_H_14_
4	10.470	1.2597	88	2,2-Dimethylbutane	86	C_6_H_14_
5	10.975	0.5609	95	3-Methylpentane	86	C_6_H_14_
6	12.95	2.1885	91	2,4-Dimethylpentane	100	C_7_H_16_
7	13.359	0.3170	95	2,2-Dimethylpentane	100	C_7_H_16_
8	14.998	0.8880	89	2-Methylhexane	100	C_7_H_16_
9	15.217	1.2700	93	2,3-Dimethylpentane	100	C_7_H_16_
10	15.578	0.6173	94	3-Methylhexane	100	C_7_H_16_
11	16.734	16.4986	90	2,2,4-Trimethylpentane	114	C_8_H_18_
12	19.879	4.4194	86	2,5-Dimethylhexane	114	C_8_H_18_
13	20.112	3.6922	88	2,4-Dimethylhexane	114	C_8_H_18_
14	20.255	2.2556	88	2,2,3-Trimethylpentane	114	C_8_H_18_
15	21.804	5.5302	89	2,3,4-Trimethylpentane	114	C_8_H_18_
16	22.407	8.5216	89	2,3,3-Trimethylpentane	114	C_8_H_18_
17	22.678	1.6362	90	2,3-Dimethylhexane	114	C_8_H_18_
18	23.130	0.6178	92	2-Methylheptane	114	C_8_H_18_
19	23.298	0.1607	88	4-Methylheptane	114	C_8_H_18_
20	23.610	0.4025	94	3,4-Dimethylhexane	114	C_8_H_18_
21	23.892	0.4454	90	3-Methylheptane	114	C_8_H_18_
22	25.049	13.3359	94	2,2,5-Trimethylhexane	128	C_9_H_20_
23	25.748	0.2054	96	2,2,4-Trimethylhexane	128	C_9_H_20_
24	27.218	0.2218	90	2,4,4-Dimethylhexane	128	C_9_H_20_
25	27.894	1.6321	87	2-Methyloctane	128	C_9_H_20_
26	28.433	0.4001	91	4,4-Dimethylheptane	128	C_9_H_20_
27	28.954	0.7304	93	2-Methyloctane	128	C_9_H_20_
28	29.530	1.3327	88	2,5-Dimethylheptane	128	C_9_H_20_
29	29.655	0.1551	88	2,2,3-Trimethylhexane	128	C_9_H_20_
30	30.606	0.0958	92	2,3,4-Trimethylhexane	128	C_9_H_20_
31	30.971	0.1409	92	1,3,5-Trimethylcyclohexane	126	C_9_H_18_
32	31.194	0.2355	90	2,3-Dimethylheptane	128	C_9_H_20_
33	31.423	0.0970	92	3,4-Dimethylheptane	128	C_9_H_20_
34	32.305	0.1495	95	2-Methyloctane	128	C_9_H_20_
35	32.665	0.3004	92	4,4-Dimethyloctane	142	C_10_H_22_
36	32.791	0.3195		unknown		C_10_H_22_
37	32.952	2.3452	87	2,2-Dimethyloctane	142	C_10_H_22_
38	33.089	0.3087	96	2,2,4-Trimethylheptane	142	C_10_H_22_
39	33.461	2.0970	87	3,4-Dimethyloctane	142	C_10_H_22_
40	34.393	1.1930	88	2,5,5-Trimethylheptane	142	C_10_H_22_
41	35.018	0.0851	87	1,1,3,5-Tetramethylcyclohexane	140	C_10_H_22_
42	35.269	0.1540	87	2,4,6-Trimethylheptane	142	C_10_H_22_
43	35.550	0.7600	90	2,3-Dimethyloctane	142	C_10_H_22_
44	35.706	0.1259	91	2,3,5-Trimethylheptane	142	C_10_H_22_
45	36.064	0.1843	87	2,5-Dimethyloctane	142	C_10_H_22_
46	36.556	0.1462	86	2,7-Dimethyloctane	142	C_10_H_22_
47	36.661	0.2365	94	3,6-Dimethyloctane	142	C_10_H_22_
48	37.957	0.0768	92	2,3-Dimethyloctane	142	C_10_H_22_
49	38.536	1.8619	90	3,8-Dimethylnonane	156	C_11_H_24_
50	39.154	0.4097	92	2,2,6,6-Tetramethylheptane	156	C_11_H_24_
51	39.320	0.7298	87	2,2,3,5-Tetramethylheptane	156	C_11_H_24_
52	39.463	0.1995	89	2,4,6-Trimethyloctane	156	C_11_H_24_
53	39.592	0.6486	88	3,6-Dimethyldecane	170	C_12_H_24_
54	39.679	0.4938	86	3-Methylundecane	170	C_12_H_24_
55	40.377	0.4704	86	3,8-Dimethyldecane	170	C_12_H_24_
56	40.503	0.1684	83	5-Methylundecane	170	C_12_H_24_
57	40.763	0.3103	81	2,8,8-Trimethyldecane	184	C_13_H_28_
58	41.564	0.1575	90	6,6-Dimethylundecane	184	C_13_H_28_
59	42.015	1.9689	86	3,9-Dimethylundecane	184	C_13_H_28_
60	42.447	0.4732	87	3,3-Dimethylundecane	184	C_13_H_28_
61	42.789	0.9040	89	3,6-Dimethylundecane	184	C_13_H_28_
62	43.579	0.3784	87	6-Methyldodecane	184	C_13_H_28_
63	43.765	0.5435	88	2,9-Dimethylundecane	184	C_13_H_28_
64	44.010	1.0890	86	5-Methyldodecane	184	C_13_H_28_
65	44.283	0.2509	90	2,5-Dimethylundecane	184	C_13_H_28_
66	44.699	1.2211	87	4,4-Dimethylundecane	184	C_13_H_28_
67	44.986	0.2573	90	5-Methyl-5-propylnonane	184	C_13_H_28_
68	45.138	0.2227	94	3-Methyldodecane	184	C_13_H_28_
69	45.311	0.2427	92	2,2,4-Trimethyldecane	184	C_13_H_28_
70	45.576	0.4157	93	2,6-Dimethylundecane	184	C_13_H_28_
71	46.537	0.2544	88	4,6-Dimethyldodecane	198	C_14_H_30_
72	46.713	0.7725	87	unknown		
73	47.060	0.1439	92	5-Methyltridecane	198	C_14_H_30_
74	47.242	0.2518	90	2,6,10-Trimethyldodecane	212	C_15_H_32_
75	48.230	0.4782	92	4-Methyltetradecane	212	C_15_H_32_
76	48.714	0.8915	90	2,6,11-Trimethyldodecane	212	C_15_H_32_
77	48.779	0.6753	90	3-Methyltetradecane	212	C_15_H_32_
78	49.045	0.2249	86	n-pentadecane	212	C_15_H_32_
79	49.355	0.1177	92	2,2-Dimethyltetradecane	226	C_16_H_34_
80	49.986	0.1678	92	n-Hexadecane	226	C_16_H_34_
81	50.404	0.2211	92	2,2,11,11-Tetramethyldodecane	226	C_16_H_34_
82	51.043	0.1408		unknown		
83	52.963	0.2317		unknown		
84	53.307	0.2980		unknown		
85	53.987	0.0631		unknown		
86	54.134	0.0924		unknown		

## Data Availability

Data is contained within article.

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
