# Peer review of "Investigation of a Complex Reaction Pathway Network of Isobutane/2-Butene Alkylation by CGC–FID and CGC-MS-DS"

_molecules, 2022, doi:10.3390/molecules27206866_

Round 1
Reviewer 1 Report
Wang reported the Investigation of a complex reaction pathway network of isobutane/2-butene alkylation by CGC–FID and CGC-MS-DS. Detail study on the alkylation process of is performed by self developed CGC–FID and CGC-MS-DS. This work provided additional informations to understand the alkylation process of sobutane/2-butene and is well designed and presented, therefore, can be accepted after minor revisions.
There are some additional comments
1) Figure, Diagram of isobutane/2-butene alkylation reaction pathway network is confusing regarding the synthesis of TMS, coupling of i-C4 with 2-C4 turned out TMPs, then why again coupling of this TMPs with i-C4 produces TMPs? Similar in case of DMHs synthesis.
2) There is no description for the DMHs, DMH stand for?
3) There are some types on the page in the section materials
Reviewer 2 Report
The manuscript entitled “Investigation of a complex reaction pathway network of isobutane/2-butene alkylation by CGC–FID and CGC-MS-DS” has been prepared and presented very well. This can be considered worth publishing in journal “Molecules” in current form.
Reviewer 3 Report
This manuscript describes a study of the acid-catalysed alkylation processes in the isobutane/2-butene system, relevant to the petrochemical industry and fuel production. The studied processes generate a very complicated hydrocarbon mixture, which has been analysed by means of CGC-MS and CGC-FID. The authors have designed and implemented their own intermittent reaction device with a metering, cooling, reaction, vacuum and analysis subsystems. With the help of this experimental setup a total of 79 volatile substances have been identified and the changes of the main chemical components in the alkylation reaction with time have been tracked. On this basis the authors propose a complex reaction network with reaction pathways between main products and byproducts. Overall, the manuscript is well written, the experimental work and the data analysis appear to be done correctly. However, I am afraid that my level of expertise in this particular field does not allow me to judge on the novelty and originality of the work. To me, this manuscript is better suited for other type of journals, such as MDPI’s “Fuels” or “ChemEngineering”. If, however, the Editor finds it suitable for Molecules, then I would suggest publication in its present form.
